# Cost-utility of cytisine for smoking cessation over and above behavioural support in people with newly diagnosed pulmonary tuberculosis: an economic evaluation of a multicentre randomised controlled trial

Jinshuo Li [1], Steve Parrott,[1] Ada Keding,[1] Omara Dogar,[1,2] Rhian Gabe,[3] Anna-Marie Marshall,[1] Rumana Huque [4,5] Deepa Barua,[5] Razia Fatima,[6] Amina Khan,[7] Raana Zahid,[7] Sonia Mansoor,[8] Daniel Kotz,[2,9,10] Melanie Boeckmann [1,9] Helen Elsey [1] Eva Kralikova,[11,12] Anne Readshaw,[1] Aziz Sheikh,[2] Kamran Siddiqi [1,13] on behalf of the TB & Tobacco Consortium

For numbered affiliations see end of article.

**Correspondence to**
Jinshuo Li;
jinshuo.li@york.ac.uk

## ABSTRACT

**Objectives** To assess the cost-effectiveness of cytisine over and above brief behavioural support (BS) for smoking cessation among patients who are newly diagnosed with pulmonary tuberculosis (TB) in low-income and middle-income countries.

**Design** An incremental cost-utility analysis was undertaken alongside a 12-month, double-blind, two-arm, individually randomised controlled trial from a public/voluntary healthcare sector perspective with the primary endpoint at 6 months post randomisation.

**Setting** Seventeen subdistrict hospitals in Bangladesh and 15 secondary care hospitals in Pakistan.

**Participants** Adults (aged ≥18 years in Bangladesh and ≥15 years in Pakistan) with pulmonary TB diagnosed within the last 4 weeks who smoked tobacco daily (n=2472).

**Interventions** Two brief BS sessions with a trained TB health worker were offered to all participants. Participants in the intervention arm (n=1239) were given cytisine (25-day course) while those in the control arm (n=1233) were given placebo. No significant difference was found between arms in 6-month abstinence.

**Primary and secondary outcome measures** Costs of cytisine and BS sessions were estimated based on research team records. TB treatment costs were estimated based on TB registry records. Additional smoking cessation and healthcare costs and EQ-5D-5L data were collected at baseline, 6-month and 12-month follow-ups. Costs were presented in purchasing power parity (PPP) adjusted US dollars (US$). Quality-adjusted life years (QALYs) were derived from the EQ-5D-5L. Incremental total costs and incremental QALYs were estimated using regressions adjusting for respective baseline values and other baseline covariates. Uncertainty was assessed using bootstrapping.

**Results** Mean total costs were PPP US$57.74 (95% CI 49.40 to 83.36) higher in the cytisine arm than in the placebo arm while the mean QALYs were −0.001 (95% CI −0.004 to 0.002) lower over 6 months. The cytisine arm was dominated by the placebo arm.

**Conclusions** Cytisine *plus* BS for smoking cessation among patients with TB was not cost-effective compared with placebo *plus* BS.

**Trial registration number** ISRCTN43811467.

## STRENGTHS AND LIMITATIONS OF THIS STUDY

⇒ Large sample size and high follow-up rate ensures robustness of the conclusion.
⇒ Comprehensive patient-level data collection provides possibilities of further exploration or updating of the analyses.
⇒ Trial across two countries posed challenges to value both costs and quality-adjusted life years comparably.
⇒ Lack of up-to-date data sources of unit costs of healthcare services may affect the accuracy of the costs estimation.
⇒ Eagerness of local staff participating in the trial may affect the generalisability of the intervention delivery.

## INTRODUCTION

In 2020, due to the impact of COVID-19 pandemic, the number of newly diagnosed tuberculosis (TB) case notifications saw a big drop from 2019 while the number of people who died from TB increased due to reduced access to services at global, regional and country levels.[1] Bangladesh (218 per 100 000 population) and Pakistan (259 per 100 000 population) are among the 16 countries that contributed most to the global shortfall

of TB notifications yet they are still on the WHO high-burden countries lists for TB and multidrug-resistant TB or rifampicin-resistant TB.[1] [2] Meanwhile, the 2020 estimates of current tobacco smoking rates were 18.5% in Bangladesh and 24.6% in Pakistan, with considerable imbalance between men and women.[3] Previous evidence suggests that continued tobacco smoking among patients with TB is associated with unfavourable TB treatment outcomes.[4] However, with the combined burden of TB and tobacco, support for smoking cessation for patients with TB is absent in both countries.[5]

TB treatment, lasting 6 months or longer, offers an opportunity for regular support for quitting smoking, if integrated properly. Newly diagnosed patients with TB who smoke might be more receptive to advice to quit due to their immediate health concerns.[6] Due to limited resources, evidence-based approaches such as behavioural support (BS) and expensive pharmacotherapies for smoking cessation cannot be implemented in many low-income and middle-income countries (LMICs). We have previously developed, in collaboration with local teams in Bangladesh and Pakistan, a brief BS integrated with routine TB appointment for smoking cessation.[7] In the present study, over-and-above the BS, we examined the effectiveness and cost-effectiveness of the relatively low cost pharmacotherapy cytisine for smoking cessation in patients with TB.[8]

We conducted a 12-month, two-arm, parallel, double-blind, placebo-controlled, multicentre, individually randomised trial in Bangladesh and Pakistan to compare cytisine plus BS for smoking cessation (cytisine arm: n=1239) with placebo plus BS (placebo arm: n=1233) among patients with pulmonary TB who smoke daily.[9] Biochemically-verified continuous abstinence at 6 months (primary endpoint) was 32.4% (401/1239) in the cytisine arm and 29.7% (366/1233) in the placebo arm (Relative Risk [RR]=1.09, 95% CI 0.97 to 1.23) and, at 12 months it was 24.9% (309/1239) and 22.3% (275/1233), respectively (RR=1.22, 95% CI 0.95 to 5.98), indicating no significant difference between arms in the primary outcome.[10] This article reports a set of analyses to, respectively: (1) evaluate the cost-utility, from a public or voluntary healthcare sector perspective, of adding cytisine to BS for smoking cessation in patients with TB who smoke; and (2) assess the financial burden in relation to tobacco use and healthcare from participants and their families' perspective, and estimate productivity loss using lost income.

## METHODS
### Design
An incremental cost-utility analysis was conducted alongside the randomised controlled trial (RCT) described above and elsewhere.[9] [10] The scheduled follow-ups were at 6 and 12 months post randomisation, with 6 months as the primary endpoint. Neither participants nor TB health workers were aware of participants' arm allocation.

Allocation was not revealed to health economists until database lock. Detailed information on procedures was provided in the study protocol.[9]

### Participants
Adults (aged ≥18 in Bangladesh and ≥15 in Pakistan) with pulmonary TB diagnosed within the last 4 weeks who smoked tobacco on a daily basis and were interested in quitting were eligible.[9] We excluded those who were diagnosed with TB complications (retreatment or any drug resistance), extrapulmonary TB, receiving streptomycin and/or para-aminosalicylic acid, using any pharmacotherapy for tobacco dependence, pregnant or planning to become pregnant, lactating or suffering from schizophrenia or known to be diagnosed with epilepsy. Those who had myocardial infarction, stroke or an attack of severe angina within the previous 2 weeks, uncontrolled high blood pressure despite being on medication or severe renal impairment (requiring dialysis) were also excluded.

Between June 2017 and April 2018, 1527 participants from 17 subdistrict hospitals in Bangladesh and 945 participants from 15 secondary care hospitals in Pakistan were randomised to the cytisine arm (n=1239) and the placebo arm (n=1233). The mean age was 42.5 (SD 14.3) years in the cytisine arm and 42.4 (SD 14.2) years in the placebo arm. Men made up 99% of each arm (1227 in the cytisine arm and 1221 in the placebo arm). By 6 months follow-up, 70 participants died (36 in the cytisine arm and 34 in the placebo arm). A further 21 participants died after 6 months (13 in the cytisine arm and 8 in the placebo arm).

### Intervention and comparator
Participants in the cytisine (intervention) arm were provided with cytisine (Desmoxan, Aflofarm, Pabianice, Poland) according to its standard regimen: 38 capsules on day 0 and another 62 capsules on day 5 (preset quit date), totalling 100 capsules over a 25-day course. The trial medication was in the form of 1.5 mg hard capsules for oral administration.[9] [10] Participants in the placebo (comparator) arm were given placebo capsules with identical appearance on the same dispensing schedule. In addition, participants in both arms were offered brief BS for smoking cessation delivered by trained TB health workers, accompanied with a leaflet containing information on tobacco use and its interactions with TB for each participant. The BS was designed to be two face-to-face sessions on days 0 (10 minutes) and 5 (5 minutes). Therefore, the intervention consisted of cytisine plus BS while the comparator was placebo plus BS.

### Measures
All monetary outcomes were collected or valued in local currencies and inflated to their respective 2018 values,[11] where necessary, and converted to purchasing power parity adjusted US dollars (PPP US$) using the World Bank exchange rate in the same year (1 PPP US$=30.9

Bangladeshi Taka=29.3 Pakistani Rupees).[12] PPP US$ accounts for the price and income difference between the two countries so that the monetary outcomes could be pooled together. Results of costs were presented in PPP US$ 2018 price.

## Costs

### Intervention costs

Intervention costs included costs of training and delivery (see online supplemental file 1). TB health workers were trained in brief BS for smoking cessation in a 2-day programme. The costs of training were estimated by the research team to be PPP US$14 183 in Bangladesh and PPP US$12 837 in Pakistan. Since all participants were scheduled to receive BS, the training cost was allocated to each participant evenly.

The uptake of BS was recorded on the case report form (CRF) on day 0. Staff costs for BS were estimated by multiplying the duration by the hourly wage rate. The cost of BS for the first and second session was PPP US$0.52 and PPP US$0.26 in Bangladesh and PPP US$0.75 and PPP US$0.38 in Pakistan. For those whose CRF showed not taking up BS, the cost of BS delivery was considered null. For those who accepted BS, the cost of the first session was applied and the cost of the second session was added provided they attended the follow-up on day 5. The smoking cessation information leaflet offered to each participant costed PPP US$0.16 in Bangladesh and PPP US$1.71 in Pakistan.

The manufacturer provided the distributor price as 72.63 Polish złoty for 100 capsule pack (PPP US$42.27 in Bangladesh and PPP US$65.09 in Pakistan). By dispensing schedule, the medication dispensed on day 0 costed PPP US$16.05 in Bangladesh and PPP US$24.74 in Pakistan, and on day 5 it costed PPP US$26.21 in Bangladesh and PPP US$40.34 in Pakistan. The placebo capsules were assumed to incur no cost. All participants had at least the first dispense and those who missed follow-up on day 5 were assumed not to receive the second dispense.

### Costs of TB treatment, additional smoking cessation help and general healthcare services

Table 1 presents the unit costs of TB treatment by phase, additional smoking cessation services and general healthcare services estimated based on secondary sources and some assumptions and converted to PPP US$ 2018[12–22] (for detailed methods of estimation see online supplemental file 1). TB treatment progression was estimated according to the TB registry card. The quantities of services use were collected by self-report at baseline, 6-month and 12-month follow-ups (see online supplemental file 2 for CRF).

### Out-of-pocket payments and productivity loss

Participants reported any spending in monetary form related to TB treatment, smoking cessation products and general healthcare services use, including travel, on CRFs at baseline, 6-month and 12-month follow-ups.

CRFs also collected participants' time spent in TB clinics and doctor visits, including travel and waiting time, and if and how many times they were accompanied by a friend or relative. The productivity loss of a companion was estimated by multiplying the overall time spent by the companion by the societal average hourly wage in the country.[20 21] We assumed that all companions were employed. Participants' productivity loss was estimated based on their self-reported duration of sick leave from work. Participants' hourly wages were extracted from secondary sources based on their occupation category and gender,[20 21] with those reported in open question reclassified according to the International Standard Classification of Occupations (online supplemental file 3, table S1).[23] Those who were unemployed, retired, students or home makers were assumed to incur no productivity loss in the case of sick leave.

### Quality-adjusted life years

The EQ-5D-5L developed by the EuroQol Group was used to measure health-related quality of life,[24] at baseline,

---

**Table 1** Unit costs of TB treatment, smoking cessation services and healthcare services

| Cost items | Unit cost (PPP US$, 2017/2018) | | Sources |
| | Bangladesh | Pakistan | |
|---|---|---|---|
| **TB treatment** | | | |
| First-line treatment, intensive phase, including drugs | 54.21 per month | 108.40 per month | 12–15 |
| First-line treatment, continuation phase, including drugs | 31.62 per month | 63.24 per month | |
| **Smoking cessation services** | | | |
| Help or advice from public/government clinic/hospital | 0.68 per use | 0.89 per use | 12 19–21 |
| Group or single counselling session at public/voluntary clinic | 0.94 per session | 1.26 per session | 12 18 20 21 |
| **General healthcare services** | | | |
| Doctor visit | 4.60 per visit | 6.83 per visit | 11 12 22 |
| Hospital inpatient | 19.06 per bed-day | 33.14 per bed-day | 11 12 22 |

PPP, purchasing power parity; TB, tuberculosis; US$, US dollars.

6-month and 12-month follow-ups, as part of the CRFs. The EQ-5D-5L consists of a descriptive system of five domains (mobility, self-care, usual activities, pain/discomfort and anxiety/depression), and a Visual Analogue Scale (VAS) valuing the overall health on the day. The VAS score ranges from 0 (death) to 100 (perfect health). Each domain of the descriptive system has five levels of capacity, ranging from having no problem to having severe problems. A complete descriptive system could be converted to a utility value using an appropriate tariff.

In the absence of country-specific valuation sets for Bangladesh and Pakistan, we used the valuation set of Zimbabwe based on crosswalk function to calculate utility,[25] as its gross domestic product per capita in PPP US\$ (2381.22) was the closest to that of the two countries of interest (Bangladesh: 4598.39 and Pakistan: 5714.03) at the time of the analysis.[26] Quality-adjusted life years (QALYs) were derived using the area under the curve approach.[27]

### Analyses

All analyses were performed using Stata/SE V.16.0.

### Missing data

For the baseline covariates, missing values were imputed by the mean of the variable in the pooled sample in the same country. This was the information that was unrelated to the intervention and the randomisation functioned to balance the two arms.[28] The missing values in the follow-up variables were handled using multiple imputation method, following Rubin's rule and assuming missing at random (MAR),[29] unless it was due to death. Missing values due to death were replaced with zero or not applicable (n/a) depending on the nature of variable. An imputation model was developed to include all the variables necessary for the analysis and the number of imputations was set as approximately the highest percentage figure of missing data.[30] The imputation was performed by trial arms and on condition of being alive.

### Primary analysis

The primary analysis was an incremental cost-utility analysis over 6 months post randomisation from a public or voluntary healthcare sector perspective. This included service providers that were classified as government, non-profit organisations and charitable organisations. It was undertaken on an intention-to-treat basis, including all randomised participants in the arms to which they were allocated.

Total costs at 6 months consisted of intervention costs, TB treatment costs, additional public/voluntary smoking cessation costs and public/voluntary healthcare services costs in the 6 months post randomisation. Mean total costs and mean QALYs were estimated for each arm and no discounting was applied for the 6 months period. Incremental mean total costs and incremental mean QALYs were estimated by a mixed effect generalised linear regression model, adjusting for their respective baseline

values (total costs in the 6 months before randomisation for total costs; baseline EQ-5D-5L utility for QALYs), age, gender and country, with study site as random-effects. An incremental cost-effectiveness ratio (ICER) was calculated by dividing the incremental mean total costs by the incremental mean QALYs.

Since there are no official willingness-to-pay (WTP) thresholds in either Bangladesh or Pakistan, the estimated WTPs for Bangladesh and Pakistan based on income elasticity of value of health, inflated to 2018 (maximum WTP: Bangladesh: PPP US\$1473 per QALY gained and Pakistan: PPP US\$2431 per QALY gained), were used to compare with the ICERs, if applicable.[31]

Because neither costs nor QALYs were normally distributed, we used a non-parametric bootstrap technique to assess the uncertainty, generating 5000 replicate samples. The results were used to construct 95% CIs of the incremental costs and QALYs. They were then plotted on a cost-effectiveness plane (CEP) to demonstrate the uncertainty surrounding the ICER. Cost-effectiveness acceptability curves (CEACs) were constructed from these bootstrapped replicates by converting ICER to net monetary benefit.[32]

A separate cost-effectiveness analysis using smoking abstinence rate at 6 months follow-up as effect measure was planned but not undertaken because no statistically significant difference was found between arms for this outcome measure per prespecified effect size.[10] Given that it is not clinically effective, it could not be cost-effective using this measure.

### Sensitivity analyses

We undertook a complete case analysis (CCA) on the participants who had complete outcome and covariates data to provide a comparison with the primary analysis based on imputed data. We examined the MAR assumption that supports the multiple imputation by undertaking sensitivity analyses based on missing not at random assumptions using a practical approximation to the pattern mixture model:[28] (1) imputed total costs were increased by 10%, 20% and 30% and (2) imputed QALYs were reduced by between 10%, 20% and 30%. To assess the impact of choice of EQ-5D-5L tariff, we took the validated population valuation sets from countries in the southeast Asia area (ie, Indonesia, Malaysia, Thailand) and the crosswalk functions of the UK and Thailand to calculate utility for comparison.[25 33–35]

### Secondary analyses

The first secondary analysis followed the methods of the primary analysis, extending time horizon to a 12-month period. No discounting was applied as this was not longer than 1 year. We summarised participants' out-of-pocket payments (OOPs) in relation to TB treatment, smoking cessation and healthcare services by arm, at both 6 and 12 months. Productivity losses of participants' sick leave and their companion to treatment and money spent on any forms of tobacco were also summarised. We have also

repeated the analysis by countries following the same methods of the primary analysis above.

## Patient and public involvement

Patient groups were consulted on the intervention materials for their lucidness during the intervention development stage. No other patient and public involvement occurred in the study process.

## RESULTS
### Missing data

The results of observed cases are presented in online supplemental file 1. The proportion of missing data at baseline was low (online supplemental file 3, table S2). The greatest percentage of missing data level was 12% of participants' OOPs for smoking cessation at 6 months follow-up, followed by the same variable at 12 months (10%).

Although the level of missingness did not differentiate between arms, most of the missingness of follow-up variables was significantly associated with country. The missingness of OOP for smoking cessation in months 1–6 was weakly associated with participants' age (online supplemental file 3, table S3). Using a logistic regression for missingness of follow-up variables on their respective previously observed values (eg, missingness of costs at 6 months on costs at baseline), most results were not statistically significant (p>0.05), with few exceptions. These results supported the MAR assumption. The imputation number was set to 15.

### Primary analysis

The mean costs of smoking cessation and healthcare services in the 6 months before baseline were PPP US$10.36 (SE PPP US$1.74) in the cytisine arm and PPP US$8.52 (SE PPP US$1.41) in the placebo arm. The mean total costs over the 6 months post randomisation were PPP US$401.52 (SE PPP US$8.91) in the cytisine arm and PPP US$334.73 (SE PPP US$5.85) in the placebo arm (table 2). Costs of additional smoking cessation were negligible in both arms. The mean costs of hospital stay in the cytisine arm were almost twice those in the placebo arm. The incremental total costs were PPP US$57.74 (95% CI PPP US$49.40 to PPP US$83.36). The mean QALYs were 0.395 (SE 0.002) in the cytisine arm and 0.398 (SE 0.002) in the placebo arm. The incremental QALYs were −0.001 (95% CI −0.004 to 0.002). The majority (78.1%, 3905/5000) of the bootstrapped replicates fell in the north-west quadrant of CEP, indicating a more costly, but less effective intervention (figure 1, left). The CEAC was not presented as it was a straight line at 0% probability of cost-effectiveness at the WTP range from PPP US$0 to PPP US$1473 per QALY gained for Bangladesh or PPP US$2431 per QALY gained for Pakistan.

### Sensitivity analyses

The CCA was performed on 1122 participants in the cytisine arm and 1116 participants in the placebo arm. The results were similar to that of the primary analysis (table 2, right). The overall majority (91%, 4550/5000) of the bootstrapped replicates fell in the north-west

**Table 2** Results of primary and complete cases analyses at 6 months post randomisation

| | Primary analysis | | Complete case analysis | |
| --- | --- | --- | --- | --- |
| | Cytisine (n=1239) | Placebo (n=1233) | Cytisine (n=1122) | Placebo (n=1116) |
| **Costs (PPP US$)** | **Mean (SE)** | | **Mean (SD)** | |
| Intervention | 60.65 (0.41) | 12.37 (0.08) | 61.25 (13.83) | 12.15 (2.69) |
| TB treatment | 305.15 (3.36) | 301.83 (3.36) | 306.53 (109.96) | 301.36 (108.09) |
| Doctor visit | 3.36 (0.37) | 3.10 (0.31) | 3.47 (13.17) | 3.14 (10.58) |
| Hospital stay | 31.91 (7.73) | 16.98 (4.41) | 33.08 (275.18) | 17.26 (151.58) |
| Smoking cessation | 0.46 (0.03) | 0.45 (0.03) | 0.49 (1.19) | 0.49 (1.13) |
| Overall total for 6 months | 401.52 (8.91) | 334.73 (5.85) | 404.82 (311.99) | 334.39 (196.52) |
| **PPP US$, mean (95% CI)** | | | | |
| Adjusted incremental costs | 57.74 (49.40 to 83.36) | | 59.49 (51.95 to 89.30) | |
| | **Mean (SE** | | **Mean (SD)** | |
| QALYs over 6 months | 0.395 (0.002) | 0.398 (0.002) | 0.401 (0.041) | 0.403 (0.039) |
| **QALYs, mean (95% CI)** | | | | |
| Adjusted incremental QALYs | −0.001 (−0.004 to 0.002) | | −0.001 (−0.003 to 0.000) | |
| **ICER** | Cytisine dominated by placebo (uncertainty, see figure 1 left) | | Cytisine dominated by placebo (uncertainty, see figure 1 right) | |

ICER, incremental cost-effectiveness ratio; PPP, purchasing power parity; QALYs, quality-adjusted life years; TB, tuberculosis; US$, US dollars.

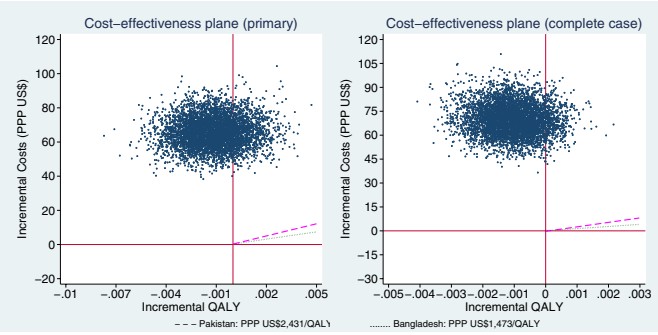

**Figure 1** Cost-effectiveness plane of primary and complete case analyses at 6 months post randomisation (dashed purple line as WTP for Pakistan and dotted green line as WTP for Bangladesh). PPP, purchasing power parity; QALY, quality-adjusted life year; US$, US dollars; WTP, willingness-to-pay.

quadrant of CEP (figure 1, right), indicating a more costly, but less effective intervention. This was consistent with the primary analysis.

Under scenario (1), when the imputed costs were increased by 10%, 20% and 30%, the incremental costs became PPP US$58.32, PPP US$58.91 and PPP US$59.51, respectively. Under scenario (2), when the imputed QALYs were reduced by 10%, 20% and 30%, the incremental QALYs were −0.001 to −0.001 and −0.000, respectively. None differed far from the primary analysis results.

Using tariffs derived in different countries or with different approaches, the incremental QALYs between arms varied (figure 2), but the level of difference was not prominent and the general pattern between arms remained the same.

### Secondary analyses

The addition of the costs in months 7–12 increased the mean total costs over 12 months to PPP US$408.31 (SE PPP US$10.03) in the cytisine arm and PPP US$341.83 (SE PPP US$6.50) in the placebo arm. The incremental costs were PPP US$56.72 (95% CI PPP US$46.58 to PPP

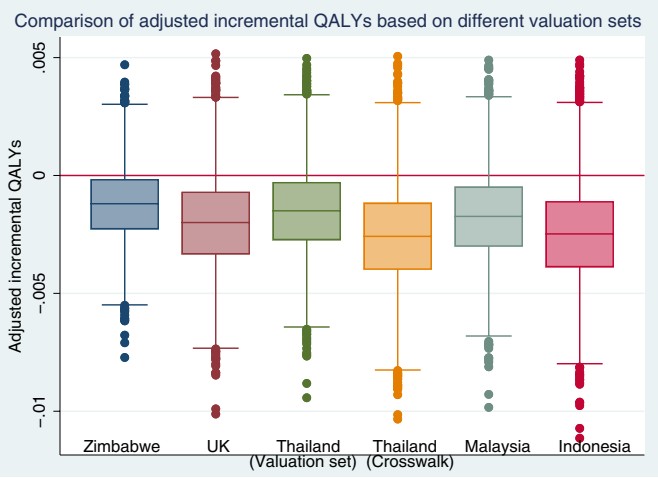

**Figure 2** Comparison of adjusted incremental QALYs over 6 months post randomisation derived from different methods. QALYs, quality-adjusted life years.

US$86.00), similar to those over the 6 months post randomisation. By contrast, as the time horizon doubled, the QALYs became almost twice as high as over the 6-month period, which led to a larger difference in mean QALYs between arms. The mean QALYs were 0.808 (SE 0.004) in the cytisine arm and 0.814 (SE 0.004) in the placebo arm. The incremental QALYs were −0.004 (95% CI −0.013 to 0.005). The cytisine arm remained dominated by the placebo arm, with 77% (4007/5000) of the bootstrapped estimates indicating a less effective, but more costly intervention.

Over the 12 months follow-up period, the mean OOPs were PPP US$108.91 (SE PPP US$19.79) in the cytisine arm and PPP US$81.74 (SE PPP US$11.73) in the placebo arm. The main cost driver was OOP for doctor visits in both arms, while in the cytisine arm participants also spent more on hospital stays (table 3). This pattern was consistent with costs from the public or voluntary healthcare sector's perspective. Productivity losses mostly occurred before and during TB treatment period and decreased considerably in the last 6 months of the trial. The OOP for tobacco products dropped after randomisation in both arms but remained stable throughout the 12 months period post randomisation, which was consistent with the quit rates observed in both arms.

The cost-utility analyses by country did not lead to different conclusions from the primary analysis. In Bangladesh, the adjusted incremental costs were PPP US$37.06 (95% CI PPP US$28.12 to PPP US$43.85) and the adjusted incremental QALYs were −0.003 (95% CI −0.006 to 0.000) with the cytisine arm remaining dominated by the placebo arm. In Pakistan, the adjusted incremental costs were PPP US$108.46 (95% CI PPP US$69.69 to PPP US$157.88) and the adjusted incremental QALYs were 0.001 (95% CI −0.004 to 0.008). The ICER was calculated at PPP US$108 464 per QALY, which was much higher than the adopted maximum WTP threshold PPP US$2431 per QALY. The cost-effectiveness plane also shows that cytisine plus BS had 0% of being cost-effective within the adopted WTP threshold range in both countries (online supplemental file 1). However, the breakdown of total costs by country indicated that the higher mean costs of hospital stay in the cytisine arm were mostly contributed by the cytisine arm in Pakistan (PPP US$78.12 vs PPP US$32.70 in placebo arm). While in Bangladesh, the mean costs of hospital stay were PPP US$3.07 (SE PPP US$1.62) in the cytisine arm and PPP US$7.34 (SE PPP US$3.82) in the placebo arm. A further examination also showed possible outliers in the cytisine arm in Pakistan. The improvement in utility from baseline to 6 months was more manifest in Bangladesh than in Pakistan, regardless of the arms. Detailed results are presented in online supplemental file 1.

### DISCUSSION

The intervention cost was PPP US$60.65 (SE PPP US$0.41) per participant in the cytisine arm and PPP US$12.37 (SE

**Table 3** Mean out-of-pocket payments for health-related services, productivity loss and payments for tobacco products at three time points, by arm

| PPP US$<br>Mean (SE) | Cytisine (n=1239) | Placebo (n=1233) |
|---|---|---|
| Six months before baseline | | |
| OOPs for health-related services | 84.90 (7.91) | 86.70 (6.80) |
| TB treatment | 15.60 (1.69) | 19.71 (3.42) |
| Doctor visit | 62.29 (6.90) | 63.96 (5.67) |
| Hospital stay | 6.97 (2.87) | 3.02 (0.80) |
| Smoking cessation | 0.04 (0.02) | 0.01 (0.01) |
| Productivity loss | 34.01 (2.14) | 30.41 (1.81) |
| OOPs for tobacco products | 1.79 (0.14) | 1.64 (0.07) |
| Months 1–6 | | |
| OOPs for health-related services | 69.70 (10.62) | 51.08 (9.32) |
| TB treatment | 22.16 (2.51) | 16.24 (1.30) |
| Doctor visit | 29.49 (7.52) | 22.65 (6.08) |
| Hospital stay | 17.65 (5.90) | 11.89 (6.53) |
| Smoking cessation | 0.40 (0.09) | 0.30 (0.06) |
| Productivity loss | 48.83 (3.00) | 43.52 (3.14) |
| OOPs for tobacco products | 0.51 (0.03) | 0.50 (0.03) |
| Months 7–12 | | |
| OOPs for health-related services | 39.21 (16.11) | 30.66 (6.72) |
| TB treatment | 5.03 (1.43) | 4.55 (0.92) |
| Doctor visit | 13.05 (2.41) | 20.42 (5.22) |
| Hospital stay | 21.08 (15.80) | 5.64 (2.89) |
| Smoking cessation | 0.04 (0.02) | 0.05 (0.02) |
| Productivity loss | 6.06 (0.58) | 8.32 (0.97) |
| OOPs for tobacco products | 0.61 (0.03) | 0.58 (0.02) |

OOPs, out-of-pocket payments; PPP, purchasing power parity; TB, tuberculosis; US$, US dollars.

PPP US$0.08) per participant in the placebo arm. The difference was mainly attributed to cytisine medication. The incremental total costs at 6 months post randomisation were estimated at PPP US$57.74 (95% CI PPP US$49.40 to PPP US$83.36) while the incremental QALYs were estimated at –0.001 (95% CI –0.004 to 0.002). These results indicated that adding cytisine to brief BS for quitting smoking was unlikely to be cost-effective. The sensitivity analyses confirmed the robustness of this conclusion. Extending the time horizon to 12 months did not change the conclusion.

While the observed quit rates were not statistically significantly different between arms,[10] participants' OOP for tobacco products on average dropped by nearly two-thirds after randomisation indicating a reduction of tobacco consumption. The higher than expected productivity loss, OOPs for doctor visits and TB treatment before baseline might be because participants had experienced some symptoms and sought medical attention before TB was diagnosed. It was unclear, however, why participants in the cytisine arm reported more and longer hospital stays than the placebo arm in Pakistan. Our process evaluation study found some difference in intervention delivery between countries,[36 37] but we did not find evidence of differential TB treatment outcomes between trial arms in Pakistan,[10] and the same situation was not observed in Bangladesh. This might indicate a potential country-related contextual reason rather than the effect of the intervention or occurrence by chance. Subgroup analyses by patient characteristics and deterministic sensitivity analysis of key parameters were not planned because of the lack of clear underlying hypotheses. Moreover, limited by the research capacity, the sample size of the subgroups was likely to be insufficient to produce valid results.

The strength of the study stems from the large sample size and high follow-up rates. Despite limitations of published data availability, patient level measures were collected using a comprehensive questionnaire to enable a full cost-utility analysis to be undertaken. However, several limitations could potentially affect the results. First, our estimated costs could be an underestimation.

We observed that some health workers discussed smoking cessation during several routine TB consultations and some research assistants delivered the study drug to participants if they had missed day 5 follow-up. TB treatment costs were estimated based on simplified scenarios. Intensive treatments in the case of deterioration, death or retreatment were not considered. Costs of general medication were not included because our unit costs data source for healthcare services did not include them. However, this should not bias the results towards either arm. Second, the data source of unit costs of healthcare services was last updated in 2010. Certain changes may not be accounted for by simple inflation. While an up-to-date data source was not available at the time of analysis, the results could be updated when it becomes available as the service use was collected in quantities. Third, productivity loss in the case of death was considered zero but if a lifetime observation or modelling were undertaken productivity loss due to premature death should be included. Given the large sample size and few deaths that occurred, this was unlikely to affect the conclusions. Last but not least, our sample consisted mostly of men. This reflected the low daily tobacco smoking rate among women in both countries at the time of the trial (0.8% in Bangladesh and 2.0% in Pakistan).[5] There may therefore be challenges in making inferences to women in these countries.

To our knowledge, this is the first cost-utility study of cytisine as a smoking cessation aid alongside an RCT and one of few for smoking cessation intervention in LMICs. A systematic review published in 2019 identified eight placebo-controlled trials and one non-inferiority trial (using nicotine replacement therapies) that used cytisine for smoking cessation, all of which were among smokers in general population and only one was conducted in LMICs.[8] Although cytisine has been identified as affordable globally,[38] its cost-effectiveness in smoking cessation was based on modelled economic evaluation not empirical evidence.[39] Our study illustrated that though less costly than other cessation aids, cytisine did not show sufficient effects to be considered cost-effective.

Our findings do not support the cost-effectiveness of adding cytisine to BS for smokers who are newly diagnosed with pulmonary TB. In the absence of more effective smoking cessation aid, future studies should explore the cost-effectiveness of non-pharmacological cessation interventions in LMICs, given the relatively lower costs of labour and possible impact of smoking-related comorbidities on quality of life in the TB population.

**Author affiliations**
[1]Department of Health Sciences, University of York, York, North Yorkshire, UK
[2]Usher Institute, The University of Edinburgh, Edinburgh, UK
[3]Centre for Cancer Prevention, Wolfson Institute of Preventive Medicine, Queen Mary University of London, London, UK
[4]Department of Economics, University of Dhaka, Dhaka, Bangladesh
[5]ARK Foundation, Dhaka, Bangladesh
[6]TB/HIV and Malaria Common Management Unit, Global Fund Grant, Islamabad, Pakistan
[7]The Initiative, Islamabad, Pakistan
[8]Institute of Psychiatry, Rawalpindi Medical University, Rawalpindi, Pakistan
[9]Institute of General Practice, Addiction Research and Clinical Epidemiology Unit, Medical Faculty, Heinrich-Heine-Universität Düsseldorf, Dusseldorf, Nordrhein-Westfalen, Germany
[10]Department of Family Medicine, CAPHRI School for Public Health and Primary Care, Maastricht University, Maastricht, The Netherlands
[11]Institute of Hygiene and Epidemiology, First Faculty of Medicine, Charles University and the General University Hospital in Prague, Praha, Czech Republic
[12]3rd Medical Department, First Faculty of Medicine, Charles University and General University Hospital in Prague, Praha, Czech Republic
[13]Hull York Medical School, University of York, York, UK

**Collaborators** TB & Tobacco Consortium (in addition to those in the main author list) Shilpi Swami, Samina Huque, Ejaz Qadeer, Maryam Noor, Salman Sohail, Jiban Karki, Iveta Nohavová, Kamila Zvolská, Alexandra Pánková, Sushil Baral (PI), Shophika Regmi, Prabin Shrestha, Basant Joshi, Ramesh Pathak.

**Contributors** JL conducted the cost-effectiveness analysis and drafted the manuscript under the supervision of SP. SP also contributed to the analysis design. AKe contributed to data management and statistical analysis, including some clinical measures used in this manuscript. OD and RG contributed to study design, conduct and interpretation of findings. AR and A-MM managed the study and contributed to interpretation of findings. RH, DB, RF, AK, RZ and SM conducted the study in Bangladesh/Pakistan, collected and managed the data in countries and provided critical inputs to data analysis and interpretation. DK, EK, MB and HE provided insights to study design on aspects of behavioural support implementation, evaluation of its delivery and interpretation of findings. AS provided critical oversight to study design, trial conduct, interpretation of findings and discussion. KS conceptualised the study, contributed to the study design, conduct and interpretation of findings. All authors provided critical revisions and approved the final manuscript. JL acts as guarantor and accepts full responsibility of the overall content.

**Funding** This work was supported by the European Union's Horizon 2020 research and innovation programme, under Grant Agreement No. 680995.

**Competing interests** KS received a research grant from Pfizer (2015–2017) to study the effects of varenicline (a smoking cessation medication) on waterpipe smoking cessation. DK received an unrestricted grant from Pfizer in 2009 for an investigator-initiated trial on the effectiveness of practice nurse counselling and varenicline for smoking cessation in primary care (Dutch Trial Register NTR3067; DOI: 10.1111/add.13927). The medication for the trial were provided by Aflofarm free of charge. AS is supported by Health Data Research UK's BREATHE Hub.

**Patient and public involvement** Patients and/or the public were involved in the design, or conduct, or reporting, or dissemination plans of this research. Refer to the Methods section for further details.

**Patient consent for publication** Not applicable.

**Ethics approval** The Health Sciences Research Governance Committee (HSRGC), University of York, UK: HSRGC/2016/144/B; The National Bioethics Committee, Pakistan Medical Research Council: no. 4–87/16/NBC-200 Part-B/RDC/4197; The National Research Ethics Committee, Bangladesh Medical Research Council: BMRC/NREC/2016–2019/1475. Participants gave informed consent to participate in the study before taking part.

**Provenance and peer review** Not commissioned; externally peer reviewed.

**Data availability statement** Data are available upon reasonable request.

**ORCID iDs**
Jinshuo Li http://orcid.org/0000-0003-1496-7450
Rumana Huque http://orcid.org/0000-0002-7616-9596
Melanie Boeckmann http://orcid.org/0000-0001-5909-5508
Helen Elsey http://orcid.org/0000-0003-4724-0581
Kamran Siddiqi http://orcid.org/0000-0003-1529-7778

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
