## [Reviewer comments · BMJ Open]

ARTICLE DETAILS

TITLE (PROVISIONAL)	Cost-utility of cytisine for smoking cessation over and above behavioural support in people with newly diagnosed pulmonary tuberculosis: An economic evaluation of a multi-centre randomised controlled trial
AUTHORS	Li, Jinshuo; Parrott, Steve; Keding, Ada; Dogar, Omara; Gabe, Rhian; Marshall, Anna-Marie; Huque, Rumana; Barua, Deepa; Fatima, Razia; Khan, Amina; Zahid, Raana; Mansoor, Sonia; Kotz, Daniel; Boeckmann, Melanie; Elsey, Helen; Kralikova, Eva; Readshaw, Anne; Sheikh, Aziz; Siddiqi, Kamran

VERSION 1 – REVIEW

REVIEWER	Trapero-Bertran, Marta Universitat Internacional de Catalunya
REVIEW RETURNED	06-Apr-2021

GENERAL COMMENTS	Dear author, Thank you very much for this interesting paper. The topic is relevant for the smoking literature and, I think, that your paper has got a value, though I would advise to address some questions before publication. Major comments 1. I am really confused with the intervention group you are using and also the groups compared. For instance, in the INTRODUCTION section you state (line 15): "...we developed a short optimised and integrated behavioural support (BS)intervention...". Then, in the METHODS section (line 19), you stated that the participant was given a leaflet and also a BS. Then, it seems that aside from this BS, the intervention group received Cytisine. So, is this drug part of the BS strategy? Are you just, then, evaluating the efficiency of this drug for quitting smoking? In order to understand appropriately this intervention, you need to review all paper and define clearly which is exactly the intervention you are evaluating.2. BS interventions always are more effective when you introduce a reminder of it after some weeks. Giving the BS, then providing some drugs, and visiting them 6 months after probably is not maximising the effectiveness of this type of interventions. Could you justify why there was no reminding or follow up in the BS intervention?3. You should justify how come that your population is mainly composed by males, and there are no females in it.4. Why you are giving the 66% of the complete treatment to the participant after 5 days? Which is the justification for that? Which is the effectiveness of this drug? Is causing any adverse events the intake of this drug to the population?
---

	5. In the INTRODUCTION section you state that the analysis is going to be conducted under the “public or voluntary sector perspective” (line 30). In line 32, you highlighted the “societal perspective”, and in the RESULTS section (line 32) you state the “healthcare provider’s perspective”. Please, could you clarify which perspective are you using? I think that both, the healthcare provider (or healthcare funder) and societal perspective, would be of interest in this analysis, and definitely presenting their results separately. 6. Please, use consistently the terminology of cost-utility analysis across the paper. Even the title uses cost-effectiveness when you are doing a cost-utility analysis. Same with the “Incremental Cost-Utility Ratio”. Please, use ICUR instead of ICER. 7. Regarding the uncertainty assessment conducted. You did a probabilistic sensitivity analysis, showing the bootstrapping results in the Cost-Effectiveness Plane. In addition, you state in the paper that you are not showing the CEAC because is a straight line in the horizontal axe. However, your plane does show some results in the first quadrant (north-east), so I think your CEAC cannot be 100% coincident with the horizontal axe. So, in your case it is worthy to show the CEAC. Moreover, probably, a deterministic sensitivity analysis of some key parameters (productivity costs) and (QALYs results) would also help understanding the uncertainty of your analysis. 8. I am definitely not an expert on TB, but I would say that depending on the severity of the TB disease you are, the benefit perception that you will have quitting smoking will be definitely underestimated. In addition, the symptoms of TB (I assume all these patients are treated for their TB disease) are treated equally in both groups, so the benefits of quitting smoking are not really perceived as a big change, because you “curing” TB anyway, and the worst symptoms for the patient (those corresponding to TB) will be disappearing. 9. I think, that comorbidities associated to smoking of these TB patients should be included and analysed, and maybe some differences on QALYs could be observed. 10. I think that DISCUSSION section needs to address in depth the following:  Why cytosine patients have longer hospital stays than placebo arm patients? Would it be possible the BS intervention in low-middle income countries work differently? Do they need to be designed differently? Would it be possible to evaluate the efficiency of this interventions by subgroups (by age, socioeconomic level, adverse events, TB severity, etc.?) You should justify why you think that this intervention is not cost-effective, and it would be nice if you could add some thoughts on which changes should suffer this intervention to become cost-effective. Minor comments  In the abstract, the results section starts presenting the “mean total costs”, but I think the relevant costs in here are the incremental mean total costs”, not really the mean total costs. In the abstracts, the conclusions section, states that “...was not cost-effective in comparison with placebo plus BS”, however I think that it would be more correct to state “...as dominated versus/compared to placebo plus BS”.
--	---

	3. Please, could you state in the paper when the costs information was collected (2017?) and, it would be nice if you could update the figures to 2020 costs. I hope these comments help. Thank you very much for this piece of work.
--	--

REVIEWER	van den Brand, Floor Maastricht University CAPHRI School for Public Health and Primary Care, Family Medicine
REVIEW RETURNED	13-Nov-2021

GENERAL COMMENTS	Review bmjopen-2021-049644 “Cost-effectiveness of cytisine for smoking cessation over and above behavioural support in people with newly diagnosed pulmonary tuberculosis: An economic evaluation of a multi-centre randomised controlled trial” This study involves a trial-based economic evaluation of adding the medication cytisine to brief smoking cessation counselling in patients who smoke with newly diagnosed tuberculosis in hospitals in Bangladesh and Pakistan. The results on the effectiveness of the intervention have been published in a different paper; the current paper involves the cost-effectiveness of the intervention. The cost-effectiveness analysis of smoking cessation medication for TB patients that was conducted in the current study is an important subject, which is under-researched in low- and middle income countries. The current study thus provides a valuable addition to the current literature. The study is well-conducted and reported according to the ISPOR CHEERS guidelines. Several sensitivity analysis that were performed increase confidence in the results. I have some comments and questions for the authors.  - The introduction is short and readers are referred to previous publications for more details on the trial. However, some extra context in the current paper may be helpful to readers. The relation between smoking and TB treatment options could be further explained. How does smoking cessation help TB patients and are benefits expected on the short- or longer term? In addition, it is not explained why Bangladesh and Pakistan are chosen to conduct the study and what was the rationale for combining the results from the two countries in the analysis. Also, I would be interested in what the standard smoking cessation care looks like for (TB) patients in the two countries and in what way it differs from the treatment that was used as an intervention in the current study. - Newly diagnosed TB patients were included in the study. Was this group chosen with a specific reason (e.g. the expectation of being more motivated to quit smoking). Do the authors think that the study results are generalizable to other TB patients (are there differences in health status)? - The study is called a cost-effectiveness analysis but when the outcome measure is QALYs, it is often referred to as a cost-
--

	utility analysis and ICUR is used instead of ICER. The authors had also planned a CEA with smoking abstinence as the primary outcome, as was written in the previously published protocol article and it probably should be mentioned in the article why the authors decided not to do this analysis.  - The data from patients in Bangladesh and Pakistan are combined in the analysis. Have the authors considered sensitivity analyses where these countries are analyzed separately? Apart from difficulties with the comparability of costs and the valuation of utilities, there may be country-related differences in environmental factors and the way treatment was delivered that may have affected how successful the treatment was and/or how it influenced quality of life. In addition, separating data may facilitate comparisons for future studies. - The study relies on the assumption that smoking cessation, possibly improved by cytosine, improves TB treatment outcomes and thereby increases quality of life. However, the potential positive effect of cytosine treatment on smoking cessation also depends on the degree that the treatment was used by patients as intended. Do the authors have information on therapy compliance and may this have influenced the results on smoking outcomes and quality of life? - There were unexplained higher costs for hospital visits in the intervention group. The authors could check whether this was caused by some extreme outliers (patients with extremely high hospital-related costs) that were unevenly distributed between the intervention and control group. - It could be clarified what comprises the public/voluntary perspective. Minor comments  - Introduction: incidence/prevalence numbers of TB and tobacco use could be added. - Methods, page 5 line 44-45: sentence may be revised - Page 16 Table 2: in the lowest row it may be clearer to state that cytosine was dominated by placebo.
--	--

VERSION 1 – AUTHOR RESPONSE

Reviewer: 1

Dr. Marta Traperó-Bertran, Universitat Internacional de Catalunya

Comments to the Author:

Major comments

1. I am really confused with the intervention group you are using and also the groups compared. For instance, in the INTRODUCTION section you state (line 15): "...we developed a short optimised and integrated behavioural support (BS)intervention...". Then, in the METHODS section (line 19), you stated that the participant was given a leaflet and also a BS. Then, it seems that aside from this BS, the intervention group received Cytisine. So, is this drug part of the BS strategy? Are you just, then, evaluating the efficiency of this drug for quitting smoking? In order to understand appropriately this intervention, you need to review all paper and define clearly which is exactly the intervention you are evaluating.

RESPONSE: Thank you for pointing out the confusion. We were evaluating the effectiveness and cost-effectiveness of the drug cytisine over and above behavioural support (BS) for quitting smoking. The BS intervention was developed in a previous study and given in both groups as usual care. The leaflet was part of BS intervention. We have now rephrased the texts and hope it would make it clearer.

CHANGES TO THE MANUSCRIPT:

Introduction:

"Due to limited resources, evidence-based approaches such as behavioural support (BS) and expensive pharmacotherapies for smoking cessation cannot be implemented in many low- and middle-income countries (LMICs). In the present study, we adopted a brief BS integrated with routine TB appointment for smoking cessation that was developed in collaboration with local teams in Bangladesh and Pakistan as part of standard usual care.[7] Over-and-above the BS we examined the effectiveness and cost-effectiveness of the relatively low cost pharmacotherapy cytisine for smoking cessation in TB patients.[8]"

Methods-Intervention and comparator:

"Participants in the cytisine (intervention) arm were provided with cytisine (Desmoxan, Aflofarm) according to its standard regimen: 38 capsules on day 0 and another 62 capsules on day 5 (pre-set quit date), totalling 100 capsules over a 25-day course. The trial medication was in the form of 1.5mg hard capsules for oral administration.[9, 10] Participants in the placebo (comparator) arm were given placebo capsules with identical appearance on the same dispensing schedule. In addition, participants in both arms were offered brief BS for smoking cessation delivered by trained TB health workers, accompanied with a leaflet containing information on tobacco use and its interactions with TB for each participant. The BS was designed to be two face-to-face sessions on days 0 (10 minutes) and 5 (5 minutes). Therefore, the intervention consisted of cytisine plus BS while the comparator was placebo plus BS."

2. BS interventions always are more effective when you introduce a reminder of it after some weeks. Giving the BS, then providing some drugs, and visiting them 6 months after probably is not maximising the effectiveness of this type of interventions. Could you justify why there was no reminding or follow up in the BS intervention?

RESPONSE: As clarified in our above response, the objectives of this study were to examine the effectiveness and cost-effectiveness of cytisine rather than BS. The brief BS thus served as

standardised usual care in our trial. Our aim was therefore not to maximise the effectiveness of BS; rather to ensure that it could be delivered in a standardised fashion and in a feasible way to all trial participants. In this pragmatic trial, consideration had to be taken of limited capacity locally. As these BS sessions were delivered by the local TB health workers during routine TB appointments, the design of the BS had to accommodate their heavy workload as they had to serve other patients as well as the participants of this trial. Keeping track of all participants and giving timely reminder required extra work that was not feasible under current circumstances. Although research assistants could provide simple reminder, this would not reflect the reality of the delivery.

3. You should justify how come that your population is mainly composed by males, and there are no females in it.

RESPONSE: Thanks. We have added the following statements in the discussion of limitations.

CHANGES TO THE MANUSCRIPT:

Discussion:

“Last but not least, our sample consisted mostly of males. This reflected the low daily tobacco smoking rates among women in both countries at the time of the trial (0.8% in Bangladesh, 2.0% in Pakistan).[5] There may therefore be challenges in making inferences to female populations.”

4. Why you are giving the 66% of the complete treatment to the participant after 5 days? Which is the justification for that? Which is the effectiveness of this drug? Is causing any adverse events the intake of this drug to the population?

RESPONSE: Sorry, we could not find anywhere mentioning 66% complete rate. The treatment completion was recorded on each participant's Case Report Form (CRF). We estimated the costs based on these records and did not 'give' percentage to any participants. The purpose of the current paper is to report cost-effectiveness results of the addition of cytisine to BS. The effectiveness of cytisine for smoking cessation was briefly reported in the Introduction: “Biochemically-verified continuous abstinence at 6 months (primary endpoint) was 32.4% (401/1239) in the cytisine arm and 29.7% (366/1233) in the placebo arm (RR=1.09, 95% CI 0.97-1.23) and at 12 months, it was 24.9% (309/1239) and 22.3% (275/1233), respectively (RR=1.22, 95% CI 0.95-5.98).” Our study did not find cytisine to be effective over and above BS in helping TB patients to quit smoking.

The relationship between any adverse events and intake of cytisine is worth investigating. We did not find difference in serious adverse events between arms and none of the events were attributed to cytisine (Dogar O, Keding A, Gabe R, et al. Cytisine for smoking cessation in patients with tuberculosis: a multicentre, randomised, double-blind, placebo-controlled phase 3 trial. *Lancet Glob Health*. 2020;8(11):e1408-e17. <https://www.ncbi.nlm.nih.gov/pubmed/33069301>.). From the costing data we gathered, there appeared to be higher costs of hospital stays in the cytisine arm than in the placebo arm. However, establishing any causal relationship is beyond the scope of this paper.

5. In the INTRODUCTION section you state that the analysis is going to be conducted under the “public or voluntary sector perspective” (line 30). In line 32, you highlighted the “societal perspective”, and in the RESULTS section (line 32) you state the “healthcare provider's perspective”. Please, could you clarify which perspective are you using? I think that both, the healthcare provider (or healthcare funder) and societal perspective, would be of interest in this analysis, and definitely presenting their results separately.

RESPONSE: Thank you for pointing this out. We would like to correct the term we used. While we used 'public or voluntary sector perspective', it was actually more specifically the “public or voluntary healthcare sector” rather than the more general meaning of this term. The 'healthcare provider's

perspective' was an error in writing. It should be 'the public or voluntary healthcare sector perspective'. We therefore changed this accordingly.

CHANGES TO THE MANUSCRIPT:

The change to "public or voluntary healthcare sector perspective" was made in Abstract-Design, Introduction-last paragraph, Methods-Analyses-Primary analysis, and Results-Secondary analyses.

RESPONSE: For the two perspectives stated in the Introduction: There are two sets of analyses, each with a different perspective. The main cost-effectiveness analysis was conducted from the public or voluntary healthcare sector perspective, which is the objective 1) in the Introduction. Objective 2) was to assess the out-of-pocket payments from participants and their families' perspective and their productivity loss due to TB treatment, which was from the societal perspective. This was phrased incorrectly and has now been rephrased.

CHANGES TO THE MANUSCRIPT:

Introduction:

"2) assess the financial burden in relation to tobacco use and healthcare from participants and their families' perspective, and productivity loss from a societal perspective".

RESPONSE: We agree that both provider's and societal perspectives are important in this analysis. We decided to only undertake the full Cost-effectiveness analysis (CEA) from public or voluntary healthcare perspective while presenting the costs from personal/family's perspective and productivity loss from societal perspective based on a few reasons. Firstly, given the trial design and data collection methods, the cost estimates from healthcare sector perspective were more accurate and reliable than those estimates of personal expenses and productivity loss. Secondly, societal costs have a much wider scope than simple productivity loss. While we attempted to address a small part of it within our study, we did not feel comfortable to present it as a full analysis as societal costs would be underestimated and the results might be misleading. Thirdly, undertaking a full CEA from a societal perspective might not be meaningful. The effectiveness measure here was QALY, which does not capture non-health effects. While it might be meaningful from healthcare or personal perspective, it is uncertain if for society as whole it could serve as a meaningful outcome measure.

6. Please, use consistently the terminology of cost-utility analysis across the paper. Even the title uses cost-effectiveness when you are doing a cost-utility analysis. Same with the "Incremental Cost-Utility Ratio". Please, use ICUR instead of ICER.

RESPONSE: Thanks for the suggestion. We would like to clarify that ICER is a measure calculated by the difference in costs divided by the difference of chosen effect measure and it is commonly used as the summary measure for both cost-effectiveness analysis and cost-utility analysis. 'Incremental Cost-Utility Ratio' is not a terminology by conventional textbooks in the field. Examples see:

<https://yhec.co.uk/glossary/incremental-cost-effectiveness-ratio-icer/> and

<https://www.gov.uk/guidance/cost-utility-analysis-health-economic-studies>

As for the term 'cost-effectiveness analysis' rather than 'cost-utility analysis', we were aware that the latter is the more accurate description of the analysis performed. However, apart from ICER mentioned above, the term 'cost-effective' or 'cost-effectiveness' are used in a few other occasions, such as 'cost-effectiveness plane', 'cost-effectiveness acceptability curves' or 'it is not cost-effective'. The term 'cost-effective/ness' is used slightly differently in these circumstances and in 'cost-effectiveness analysis'. There is no terminology as 'cost-utility plane' or 'cost-utility acceptability curves', and neither would it be appropriate to make a statement such as 'it is not cost-utilisable'. In order to make the article more consistent and avoid confusion for the readers that are not familiar with the terms in health economics, we used 'cost-effectiveness analysis' as an umbrella term to describe

the cost-utility analysis conducted. To make it more accurate, we have now changed the to 'cost-utility analysis' where applicable.

CHANGES TO THE MANUSCRIPT:

The change to 'cost-utility' has been applied to the title, Abstract-Design, Introduction-last paragraph, Methods-Design, Methods-Analyses-Primary analysis, and Discussion.

7. Regarding the uncertainty assessment conducted. You did a probabilistic sensitivity analysis, showing the bootstrapping results in the Cost-Effectiveness Plane. In addition, you state in the paper that you are not showing the CEAC because is a straight line in the horizontal axe. However, your plane does show some results in the first quadrant (north-east), so I think your CEAC cannot be 100% coincident with the horizontal axe. So, in your case it is worthy to show the CEAC. Moreover, probably, a deterministic sensitivity analysis of some key parameters (productivity costs) and (QALYs results) would also help understanding the uncertainty of your analysis.

RESPONSE: Thanks for the suggestion regarding Probabilistic analysis. However, we would like to note that bootstrapping method is a non-parametric resampling technique, not a form of probabilistic sensitivity analysis. A probabilistic sensitivity analysis requires pre-assigned distributions to the parameters and then resampling values for each parameter via their individual distribution. We do not believe that bootstrap based sensitivity analysis could be classified as probabilistic.

Thanks for pointing out the issue of the straight line on the CEAC. We missed the willingness-to-pay (WTP) threshold range in the paper. While there were some results falling in the first quadrant, they would only be considered cost-effective if they were below the adopted WTP threshold. In this case, none of them were below the WTP thresholds adopted. It should be noted that the CEAC is a straight line within the range from 0 to the maximum WTP threshold. We have now added the range in the text and threshold lines in the figure of CEA.

CHANGES TO THE MANUSCRIPT:

Methods-Analyses-Primary analysis:

"Since there are no official willingness-to-pay (WTP) thresholds in either Bangladesh or Pakistan, the estimated WTPs for Bangladesh and Pakistan based on income elasticity of value of health, inflated to 2018 (maximum WTP: Bangladesh: PPP US\$1,473 per QALY gained; Pakistan: PPP US\$2,431 per QALY gained), were used to compare with the ICERs, if applicable.[31]"

Results-Primary analysis:

"The CEAC was not presented as it was a straight line at 0% probability of cost-effectiveness at the WTP range from PPP US\$0 to PPP US\$1,473 per QALY gained for Bangladesh or PPP US\$2,431 per QALY gained for Pakistan."

Figure 1 has been adjusted by adding maximum WTP thresholds.

RESPONSE: We appreciate that further investigation into the uncertainty might be worthwhile. However, without any theoretical support, it might not be appropriate to arbitrarily change values of parameters. The value ranges chosen for deterministic sensitivity analysis should have been determined in the analysis design stage with sufficient literature or empirical evidence. We would consider this for future studies but in the present article, we do not believe that it is appropriate to add analyses that were not in the approved analysis plan. We have added the following statement in the discussion to address this point.

CHANGES TO THE MANUSCRIPT:

Discussion:

“Lack of any clear underlying hypotheses or evidence, subgroup analyses by patient characteristics or deterministic sensitivity analysis of key parameters were not planned and the sample size concerning these factors was likely to be insufficient.”

8. I am definitely not an expert on TB, but I would say that depending on the severity of the TB disease you are, the benefit perception that you will have quitting smoking will be definitely underestimated. In addition, the symptoms of TB (I assume all these patients are treated for their TB disease) are treated equally in both groups, so the benefits of quitting smoking are not really perceived as a big change, because you “curing” TB anyway, and the worst symptoms for the patient (those corresponding to TB) will be disappearing.

RESPONSE: Yes, we agree that the benefit perception of quitting smoking could be underestimated when TB treatment is in progress. We used QALYs to measure participants’ perception of their health as a whole. It would not be feasible to ask participants to separate the perceived benefits of quitting smoking and TB treatment. We also agree that comparing to the TB symptoms, any changes due to quitting smoking would appear negligible. The rationale of our study was based on evidence that quitting smoking at the beginning of TB treatment could improve TB treatment outcome, e.g. increasing the cure rate. The benefits, if present, would be reflected by the general health condition. As reported by another publication from our trial, when comparing quitters and non-quitters, benefits were observed for quitters in some TB-related outcomes (Siddiqi K, Keding A, Marshall AM, et al. Effect of quitting smoking on health outcomes during treatment for tuberculosis: secondary analysis of the TB & Tobacco Trial. *Thorax*. 2021. <https://www.ncbi.nlm.nih.gov/pubmed/34272336>).

9. I think, that comorbidities associated to smoking of these TB patients should be included and analysed, and maybe some differences on QALYs could be observed.

RESPONSE: Thank you for the suggestion. We agree that this might be of interest in future studies. However, as we had not planned to analyse comorbidities, we did not collect the relevant data and therefore could not perform such analysis. Our study recruited participants from routine TB clinics, and because we intended to test a drug, people with certain cardiovascular illnesses were excluded from the trial. Both limited the number of participants with comorbidities of interest, which might lead to insufficient data for analysis of comorbidities in any case. We have now added this as a future direction of research.

CHANGES TO THE MANUSCRIPT:

Discussion:

“Our findings do not support the cost-effectiveness of adding cytosine to BS for smokers who are newly diagnosed with pulmonary TB. In the absence of more effective smoking cessation aid, future studies should explore the cost-effectiveness of non-pharmacological cessation interventions in LMICs, given the relatively lower costs of labour, and possible impact of smoking-related comorbidities on quality of life in the TB population.”

10. I think that DISCUSSION section needs to address in depth the following:
a. Why cytosine patients have longer hospital stays than placebo arm patients?

RESPONSE: This is an issue that puzzled us, too. We have raised this issue in the Discussion but we do not have data to try to explain the reason behind it. In response to the comment 4 made by reviewer 2 Dr. Floor van den Brand, we added the results by country, which show that the longer hospital stay in cytosine arm only occurred in Pakistan and might be resulted from some outliers in the cytosine arm. This might suggest that it is associated with country specific factors rather than the intervention, or it might have occurred by chance. We have now added the following to the Discussion.

CHANGES TO THE MANUSCRIPT:

Discussion:

“It was unclear, however, why participants in the cytosine arm reported more and longer hospital stays than the placebo arm in Pakistan. Our process evaluation study found some difference in intervention delivery between countries,[36, 37] but we did not find evidence of differential TB treatment outcomes between trial arms in Pakistan,[10] and the same situation was not observed in Bangladesh. This might indicate a potential country-related contextual reason rather than the effect of the intervention, or occurrence by chance.”

b. Would it be possible the BS intervention in low-middle income countries work differently? Do they need to be designed differently?

RESPONSE: This BS intervention was customised for use in low- and middle-income countries so this should not be an issue. We have added this clarification in the Introduction.

CHANGES TO THE MANUSCRIPT:

Introduction:

“In the present study, we adopted a brief BS integrated with routine TB appointment for smoking cessation that was developed in collaboration with local teams in Bangladesh and Pakistan as part of standard usual care.[7]”

c. Would it be possible to evaluate the efficiency of this interventions by subgroups (by age, socioeconomic level, adverse events, TB severity, etc.?)

RESPONSE: Thanks for the suggestion. Subgroups might be a factor in the efficiency of this intervention. However, we do not have any clear underlying hypotheses to be tested in this respect. Furthermore, as these were not planned, they would be post-hoc exploratory analyses and likely to be underpowered to draw any conclusion.

CHANGES TO THE MANUSCRIPT:

Discussion:

“Lack of any clear underlying hypotheses or evidence, subgroup analyses by patient characteristics or deterministic sensitivity analysis of key parameters were not planned and the sample size concerning these factors was likely to be insufficient.”

d. You should justify why you think that this intervention is not cost-effective, and it would be nice if you could add some thoughts on which changes should suffer this intervention to become cost-effective.

RESPONSE: As the response to point 7 above, we concluded that the intervention is not cost-effective as the probability of cost-effective was zero within the adopted WTP thresholds. The majority of the estimates fell into the more costly but less effective quadrant and the small portion that were in the more costly and more effective quadrant were all above WTP thresholds. Therefore, we concluded that the intervention was not cost-effective.

CHANGES TO THE MANUSCRIPT:

Discussion:

“Our study illustrated that though less costly than other cessation aids, cytosine did not show sufficient effects to be considered cost-effective.”

RESPONSE: For the changes to be made to make this intervention cost-effective, we have attempted

to indicate this issue in our final conclusion: “Future studies might explore non-medical interventions in LMICs, given the relatively lower costs of labour.” But reading it again, we realised that this was not clear. The reason behind the intervention not being cost-effective was that cytisine did not demonstrate significant effect in quitting smoking while adding on to the costs, over and above brief BS. As cytisine is a drug with standard regime, we do not believe that any changes could be made to it from our end to affect its effects. Pharmacological scientists might be able to guide possible changes but it is beyond the scope of this paper. On the other hand, because the comparator arm received placebo and we did not have a third arm receiving brief BS only, we could not be sure that there was no placebo effect added on to the effects of BS. Given that it would not be ethical to give placebo in practice or conduct a trial to compare brief BS with nothing, it might be a good idea to establish brief BS for smoking cessation in routine care in some areas first and compare the quit rate in these areas with those with no such service established.

CHANGES TO THE MANUSCRIPT:

Discussion:

“Our findings do not support the cost-effectiveness of adding cytisine to BS for smokers who are newly diagnosed with pulmonary TB. In the absence of more effective smoking cessation aid, future studies should explore the cost-effectiveness of non-pharmacological cessation interventions in LMICs, given the relatively lower costs of labour, and possible impact of smoking-related comorbidities on quality of life in the TB population.”

Minor comments

1. In the abstract, the results section starts presenting the “mean total costs”, but I think the relevant costs in here are the incremental mean total costs”, not really the mean total costs.

RESPONSE: The full sentence is “Mean total costs were PPP US\$57.74 (95% CI 49.40 – 83.36) higher in the cytisine arm than in the placebo arm”. We used a comparative form to make clear the direction of the incremental mean total costs, so the ‘mean total costs’ as used here is the correct term.

2. In the abstracts, the conclusions section, states that “...was not cost-effective in comparison with placebo plus BS”, however I think that it would be more correct to state “...as dominated versus/compared to placebo plus BS”.

RESPONSE: The ‘intervention dominated by control’ is one scenario of ‘intervention being not cost-effective’ so it is a more precise statement. We have now changed it accordingly.

CHANGES TO THE MANUSCRIPT:

Abstract-Conclusions:

“Cytisine plus BS for smoking cessation among TB patients was dominated by placebo plus BS.”

3. Please, could you state in the paper when the costs information was collected (2017?) and, it would be nice if you could update the figures to 2020 costs.

RESPONSE: Thank you for pointing this out. We have stated the all costs were inflated to 2018 values but did not make it clear enough. We have now rephrased the statement.

CHANGES TO THE MANUSCRIPT:

Design-Measures:

“All monetary outcomes were collected or valued in local currencies and inflated to their respective 2018 values [11] where necessary, and converted to purchasing power parity adjusted US dollars (PPP US\$) using the World Bank exchange rate in the same year (1 PPPUS\$ = 30.9 Bangladeshi

Taka = 29.3 Pakistani Rupees).[12] PPP US\$ accounts for the price and income difference between the two countries so that the monetary outcomes could be pooled together. Results of costs were presented in PPP US\$ 2018 price.”

RESPONSE: In the Methods section, we stated that the participants were randomised between June 2017 and April 2018. With 12 months follow-up period, the data were collected from 2017 to 2019. We chose the middle year as the price year. As the trial was conducted during 2017-2019, we do not feel comfortable to inflate cost to 2020 price. Firstly, as mentioned in the Discussion, the unit costs of healthcare services have already been inflated from 2010 and to inflate more might result in more uncertainty. Secondly, things have changed a lot in the past two years; the prices updated to 2020 would not reflect the reality when the trial conducted. Since this is an economic evaluation alongside a randomised controlled trial, we believe it to be better kept consistent with the trial settings and time. It might be more appropriate for a systematic review or modelling exercises to undertake further adjustment.

Thank you for your careful reading of our paper and your constructive critique of our work.

=====

Reviewer: 2

Dr. Floor van den Brand, Maastricht University CAPHRI School for Public Health and Primary Care
Comments to the Author:

1- The introduction is short and readers are referred to previous publications for more details on the trial. However, some extra context in the current paper may be helpful to readers. The relation between smoking and TB treatment options could be further explained. How does smoking cessation help TB patients and are benefits expected on the short- or longer term? In addition, it is not explained why Bangladesh and Pakistan are chosen to conduct the study and what was the rationale for combining the results from the two countries in the analysis. Also, I would be interested in what the standard smoking cessation care looks like for (TB) patients in the two countries and in what way it differs from the treatment that was used as an intervention in the current study.

RESPONSE: Thank you for the suggestion. In our original submission, we ended up removing large parts of the Introduction in order to comply with word constraints. We have reworked aspects of the paper to free up space to provide a fuller description of the context for this work.

CHANGES TO THE MANUSCRIPT:

Introduction:

“In 2020, due to the impact of COVID-19 pandemic, the number of newly diagnosed tuberculosis (TB) case notifications saw a big drop from 2019 while the number of people who died from TB increased at global, regional, and country levels.[1] Bangladesh (218 per 100,00 population) and Pakistan (259 per 100,000 population) are among the 16 countries that contributed most to the global shortfall of TB notifications yet they are still on the World Health Organization high-burden countries lists for TB and multidrug-resistant TB or rifampicin-resistant TB.[1,2] Meanwhile, the 2020 estimates of current tobacco smoking rates were 18.5% in Bangladesh and 24.6% in Pakistan, with considerable imbalance between males and females.[3] Previous evidence suggests that continued tobacco smoking among TB patients is associated with unfavourable TB treatment outcomes.[4] However, with the combined burden of TB and tobacco, support for smoking cessation for TB patients is absent in both countries.[5]”

RESPONSE: For the rationale of combining Bangladesh and Pakistan, apart from what was mentioned above regarding the general situation of TB and smoking in both countries, we chose Bangladesh and Pakistan for this study also because of practical reasons. The brief BS used in the study was developed in collaboration with local teams in the countries in the previous stage of our

research programme, so it did not require additional adaptation to fit the local settings. Our research team also built a good working relationship with local national tuberculosis programme, which made this large trial feasible.

We decided a priori to combine the sample in the two countries for pragmatic reasons as a sufficiently large sample size was needed to produce a clinically meaningful difference between arms. In addition, the main purpose was to assess the effectiveness and cost-effectiveness of cytosine for smoking cessation. The brief BS only served as a common ground for all participants. Since cytosine is a drug, we did not expect that it would have different effects on people from different countries.

Smoking cessation care was absent for the general population at the time of the trial, let alone anything standard for TB patients. This was one of the reasons behind our research programme. In the earlier stage of the programme, our research team developed with the collaboration of local teams a brief BS that is considered feasible to be delivered in routine TB treatment appointment. This trial was intended as a step further and assess if the evidence-based approach of BS + pharmacotherapies would be effective in LMICs as in developed countries such as the UK.

CHANGES TO THE MANUSCRIPT

Introduction:

“TB treatment, lasting six months or longer, offers an opportunity for regular support for quitting smoking, if integrated properly. Newly diagnosed TB patients who smoke might be more receptive to advice to quit due to their immediate health concerns.[6] Due to limited resources, evidence-based approaches such as behavioural support (BS) and expensive pharmacotherapies for smoking cessation cannot be implemented in many low- and middle-income countries (LMICs). In the present study, we adopted a brief BS integrated with routine TB appointment for smoking cessation that was developed in collaboration with local teams in Bangladesh and Pakistan as part of standard usual care.[7] Over-and-above the BS, we examined the effectiveness and cost-effectiveness of the relatively low cost pharmacotherapy cytosine for smoking cessation in TB patients.[8]”

2- Newly diagnosed TB patients were included in the study. Was this group chosen with a specific reason (e.g. the expectation of being more motivated to quit smoking). Do the authors think that the study results are generalizable to other TB patients (are there differences in health status)?

RESPONSE: There were several reasons for choosing this patient population. Firstly, newly diagnosed TB patients are the majority of the TB patients, which helped with the recruitment. Secondly, offering smoking cessation support at the time when TB is diagnosed (as opposed to any other point in their clinical pathway) was considered as the most practical, acceptable and likely to be most beneficial entry point. This was stated in the Introduction: “TB treatment, lasting six months or longer, offers an opportunity for regular support for quitting smoking, if integrated properly. Newly diagnosed TB patients who smoke might be more receptive to advice to quit due to their immediate health concerns”.

As for the generalisability, TB patients with other complications, such as drug-resistance, relapses and so on, are usually in a worse condition and take specific medicines to treat their condition, some of which interacts with cytosine, e.g. streptomycin. The suitability of cytosine for TB patients of complicated health state requires extra examination and the interaction of cytosine and other types of medications has to be studied carefully. Therefore, we do not think that the results are generalizable to other TB patients, not only because different health status, but also some of them could not take cytosine.

3- The study is called a cost-effectiveness analysis but when the outcome measure is QALYs, it is often referred to as a cost-utility analysis and ICUR is used instead of ICER. The authors had also

planned a CEA with smoking abstinence as the primary outcome, as was written in the previously published protocol article and it probably should be mentioned in the article why the authors decided not to do this analysis.

RESPONSE: Thanks for the suggestion. We have now changed the term 'cost-effectiveness analysis' to 'cost-utility analysis'. However, we would like to clarify that ICER is a measure calculated by the difference in costs divided by the difference of chosen effect measure and it is commonly used as the summary measure for both cost-effectiveness analysis and cost-utility analysis. 'Incremental Cost-Utility Ratio' is not a terminology by convention in the field. Examples see: <https://yhec.co.uk/glossary/incremental-cost-effectiveness-ratio-icer/> and <https://www.gov.uk/guidance/cost-utility-analysis-health-economic-studies> Also see point 6 of Reviewer 1 Dr. Marta Trapero-Bertran.

CHANGES TO THE MANUSCRIPT:

The change to 'cost-utility' has been applied to the title, Abstract-Design, Introduction-last paragraph, Methods-Design, Methods-Analyses-Primary analysis, and Discussion.

RESPONSE: Yes, it is correct that we initially planned a CEA using smoking abstinence. However, after analysing the quit rates following the analysis plan, there was no statistically significant difference found between arms. It was therefore concluded that cytisine + BS was not more effective in terms of smoking abstinence than placebo + BS [described in Introduction: Biochemically-verified continuous abstinence at 6 months (primary endpoint) was 32.4% (401/1239) in the cytisine arm and 29.7% (366/1233) in the placebo arm (RR=1.09, 95% CI 0.97-1.23)]. Given that the intervention was not clinically effective in terms of abstinence, it could not be cost-effective, regardless of the costs. We therefore did not undertake the CEA using smoking abstinence as planned. We do apologize for neglecting to state this in the article. We have now added a statement to explain why CEA with smoking abstinence was not performed.

CHANGES TO THE MANUSCRIPT

Methods-Analyses-Primary analysis:

"A separate cost-effectiveness analysis using smoking abstinence rate at six months follow-up as effect measure was planned but not undertaken because no statistically significant difference was found between arms for this outcome measure per pre-specified effect size.[9] Given that it is not clinically effective, it could not be cost-effective using this measure."

4- The data from patients in Bangladesh and Pakistan are combined in the analysis. Have the authors considered sensitivity analyses where these countries are analyzed separately? Apart from difficulties with the comparability of costs and the valuation of utilities, there may be country-related differences in environmental factors and the way treatment was delivered that may have affected how successful the treatment was and/or how it influenced quality of life. In addition, separating data may facilitate comparisons for future studies.

RESPONSE: To assess the cost-effectiveness of cytisine, when estimating the incremental costs and QALYs, the study site was included as a random effects factor to account for possible contextual difference due to countries and site areas. In fact, we undertook another set of analyses (primary + sensitivity) by country separately. Because of the limited space of a paper and the objective of the trial to assess difference between arms rather than by country, we only presented the overall results, though the results by country did not differ much. We have now added a brief summary of the results by country to secondary analysis and details to the supplementary file for those who are interested.

CHANGES TO THE MANUSCRIPT

Methods-Analyses-Secondary analyses:

“We have also repeated the analysis by countries following the same methods of the primary analysis above.”

Results-Secondary analyses:

“By country analyses did not lead to different conclusions from the primary analysis. In Bangladesh, the adjusted incremental costs were PPP US\$37.06 (95% CI PPP US\$28.12 to PPP US\$43.85) and the adjusted incremental QALYs were -0.003 (95% CI -0.006 to 0.000), with the cytosine arm remaining dominated by the placebo arm. In Pakistan, the adjusted incremental costs were PPP US\$108.46 (95% CI PPP US\$69.69 to PPP US\$157.88) and the adjusted incremental QALYs were 0.001 (95% CI -0.004 to 0.008). The ICER was calculated at PPP US\$108,464 per QALY, which was much higher than the adopted maximum WTP threshold PPP US\$2,431 per QALY. The cost-effectiveness plane also shows that cytosine plus BS had 0% of being cost-effective within the adopted WTP threshold range in both countries (Supplementary information 1). However, the breakdown of total costs by country indicated that the higher mean costs of hospital stay in the cytosine arm were mostly contributed by the cytosine arm in Pakistan (PPP US\$78.12 vs PPP US\$32.70 in placebo arm). While in Bangladesh, the mean costs of hospital stay were PPP US\$3.07 (SE PPP US\$1.62) in the cytosine arm and PPP US\$7.34 (SE PPP US\$3.82) in the placebo arm. A further examination also showed possible outliers in the cytosine arm in Pakistan. The improvement in utility from baseline to six months was more manifest in Bangladesh than in Pakistan, regardless of the arms. Detailed results are presented in Supplementary information 1.”

5- The study relies on the assumption that smoking cessation, possibly improved by cytosine, improves TB treatment outcomes and thereby increases quality of life. However, the potential positive effect of cytosine treatment on smoking cessation also depends on the degree that the treatment was used by patients as intended. Do the authors have information on therapy compliance and may this have influenced the results on smoking outcomes and quality of life?

RESPONSE: Yes, we collected data on self-reported therapy compliance, which was reported in the clinical effectiveness paper published earlier (Dogar O, Keding A, Gabe R, et al. Cytisine for smoking cessation in patients with tuberculosis: a multicentre, randomised, double-blind, placebo-controlled phase 3 trial. *Lancet Glob Health*. 2020;8(11):e1408-e17. <https://www.ncbi.nlm.nih.gov/pubmed/33069301>). Self-reported medication compliance was high (>90%) and similar in both treatment groups. While it might have an influence on smoking outcomes and quality of life, our trial was a pragmatic one, which reflected what could happen in reality. Patients do not always follow instructions, and that would be reflected in our results.

6- There were unexplained higher costs for hospital visits in the intervention group. The authors could check whether this was caused by some extreme outliers (patients with extremely high hospital-related costs) that were unevenly distributed between the intervention and control group.

RESPONSE: Please also see point 10a by Reviewer 1 Dr. Marta Trapero-Bertran. After adding by country analysis, in response to the comment 4 above, it shows that the higher costs of hospital stay in the cytosine arm were only the case in Pakistan, not in Bangladesh. Overall, the costs of hospital stay were much higher in Pakistan than in Bangladesh. We have followed the suggestion and checked outliers. It shows that there were possible outliers in the cytosine arm in Pakistan but not in the placebo arm or in Bangladesh. However, even without the possible outliers, participants in the cytosine arm in Pakistan still reported slightly higher costs of hospital stay than those in the placebo arm. Due to limited space of the main body of the paper, we have added only a brief statement of this to the text (see by country analysis above) but added more details in supplementary information 1. Related statements have also been added in discussion.

CHANGES TO THE MANUSCRIPT:

Discussion:

“It was unclear, however, why participants in the cytosine arm reported more and longer hospital stays than the placebo arm in Pakistan. Our process evaluation study found some difference in intervention delivery between countries,[36, 37] but we did not find evidence of differential TB treatment outcomes between trial arms in Pakistan,[10] and the same situation was not observed in Bangladesh. This might indicate a potential country-related contextual reason rather than the effect of the intervention, or occurrence by chance.”

7- It could be clarified what comprises the public/voluntary perspective.

RESPONSE: Thanks for pointing this out. First, we now changed the ‘public/voluntary perspective’ to ‘public/voluntary healthcare perspective’. We added ‘healthcare’ to make the perspective taken clearer. We also added the sentence to clarify what comprises the public/voluntary healthcare perspective.

CHANGES TO THE MANUSCRIPT

Methods-Analyses-Primary analysis:

“This included service providers that were classified as government, non-profit organisations, and charitable organisations.” to the description of primary analysis in methods section.

Minor comments

8- Introduction: incidence/prevalence numbers of TB and tobacco use could be added.

RESPONSE: Thanks for the suggestion. These have now been added.

CHANGES TO THE MANUSCRIPT

Introduction:

“Bangladesh (218 per 100,00 population) and Pakistan (259 per 100,000 population) are among the 16 countries that contributed most to the global shortfall of TB notifications yet they are still on the World Health Organization high-burden countries lists for TB and multidrug-resistant TB or rifampicin-resistant TB.[1, 2] Meanwhile, the 2020 estimates of current tobacco smoking rates were 18.5% in Bangladesh and 24.6% in Pakistan, with considerable imbalance between male and female.[3]”

9- Methods, page 5 line 44-45: sentence may be revised

RESPONSE: Sorry, we are unsure if we have found the correct sentence. The sentences we found on page 5 line 44-45 was “Allocation was not revealed to health economists until database lock. Detailed information please see study protocol attached as supplementary file.” Please let us know if this is not what was referred to.

CHANGES TO THE MANUSCRIPT

“Allocation was not revealed to health economists until database lock. Detailed information on procedures was provided in study protocol.[9]”

10- Page 16 Table 2: in the lowest row it may be clearer to state that cytosine was dominated by placebo.

RESPONSE: Thanks. It has now been changed in the table.

Thanks you for taking time reading our article and providing valuable comments.

VERSION 2 – REVIEW

REVIEWER	van den Brand, Floor Maastricht University CAPHRI School for Public Health and Primary Care, Family Medicine
REVIEW RETURNED	03-Jan-2022

GENERAL COMMENTS	Second Review bmjopen-2021-049644 “Cost-effectiveness of cytisine for smoking cessation over and above behavioural support in people with newly diagnosed pulmonary tuberculosis: An economic evaluation of a multi-centre randomised controlled trial” The authors have addressed my comments satisfactorily and have made good improvements to the manuscript. Some additional comments based on: Main Document – Marked copy (revised):  - Introduction, line 38-40: the authors second aim was to: “assess the financial burden in relation to tobacco use and healthcare from participants and their families’ perspective, and productivity loss from a societal perspective.” It is not entirely clear to me which analyses were performed to answer this research question. If I understand correctly, no cost-utility analysis was performed from a societal or patient perspective. With the data collected, it is possible to perform a cost-utility analysis from a societal perspective where treatment/healthcare costs, patient costs and costs of productivity loss are combined. Was there a reason to focus on the public or voluntary healthcare sector perspective? - Table 2 may be simplified/visually improved. Some of the grey horizontal cells and bold font/italic font are a bit distracting and may not be necessary. - Introduction, line 3: “while the number of people who died from TB increased” - Introduction, line 9: “with considerable imbalance between male and female”. The authors may consider ‘men and women’ (also in Discussion paragraph). - Introduction, line 20-22: “In the present study, we adopted a brief BS integrated with routine TB appointment for smoking cessation....” Authors may consider revising this sentence. - Discussion, page 11, line 1: “Lack of any clear underlying hypotheses or evidence, subgroup analyses by patient characteristics or deterministic sensitivity analysis of key parameters were not planned..”. Authors may consider revising this sentence. - Supporting information_1 (detailed methods and results of secondary analyses: “Upon further investigation, there were more participants who had hospital stays in Pakistan than in Bangladesh in both arms”. In addition, Figure 2 has two values with only 2 decimals instead of 3.
--

VERSION 2 – AUTHOR RESPONSE

Reviewer: 2

Dr. Floor van den Brand, Maastricht University CAPHRI School for Public Health and Primary Care
Comments to the Author:

Second Review bmjopen-2021-049644

“Cost-effectiveness of cytisine for smoking cessation over and above behavioural support in people with newly diagnosed pulmonary tuberculosis: An economic evaluation of a multi-centre randomised controlled trial”

The authors have addressed my comments satisfactorily and have made good improvements to the manuscript.

Thank you for the previous constructive comments. It is much appreciated.

Some additional comments based on: Main Document – Marked copy (revised):

- Introduction, line 38-40: the authors second aim was to: “assess the financial burden in relation to tobacco use and healthcare from participants and their families’ perspective, and productivity loss from a societal perspective.” It is not entirely clear to me which analyses were performed to answer this research question. If I understand correctly, no cost-utility analysis was performed from a societal or patient perspective. With the data collected, it is possible to perform a cost-utility analysis from a societal perspective where treatment/healthcare costs, patient costs and costs of productivity loss are combined. Was there a reason to focus on the public or voluntary healthcare sector perspective?

Thank you for pointing this out. We presented patient costs and productivity loss to assess the financial burden from participants and their families’ perspective. We considered the current data insufficient to reflect the societal perspective. The first reason was that the productivity loss in the study was measured by lost income. While it is one way to measure productivity loss, we felt that, for it to be used in a cost-utility analysis, more factors have to be considered. For instance, if the patients taking paid leave or unpaid leave or anything between unpaid and 100% paid, if replacement are hired, if production is in fact delayed/reduced due to their sickness, and so on. Patients’ absence from work do not necessarily translate in full to societal production loss. In its current form, we presented it as an rough estimate to reflect productivity loss but did not feel confident to use it in a formal analysis. The second reason was that the utility was measured in the form of health-related quality of life, which was more relevant to the healthcare sector perspective. This effect measure is not completely compatible with societal perspective. The third reason was that the costs of the intervention (cytisine), in the case of being effective, would have been borne by the public or voluntary healthcare sector. Private sectors, such as employers, are unlikely to pay for their employees to quit smoking. We believe that research should be focused on the aspects that are more pertinent to the policy making and key stakeholders.

We could see why this could cause confusion. We have removed the ‘societal perspective’ and changed the aim 2) to:

“2) assess the financial burden in relation to tobacco use and healthcare from participants and their families’ perspective, and estimate productivity loss using lost income.”

- Table 2 may be simplified/visually improved. Some of the grey horizontal cells and bold font/italic font are a bit distracting and may not be necessary.

Thanks for the suggestion. We have removed grey shadings and italic font and only left bold font for the headings.

- Introduction, line 3: “while the number of people who died from TB increased”

This is correct. We have added clarification:

“while the number of people died from TB increased due to reduced access to services at global, regional, and country levels.”

- Introduction, line 9: “with considerable imbalance between male and female”. The authors may consider ‘men and women’ (also in Discussion paragraph).

Thanks for the suggestion. We have now changed the term ‘male’ and ‘female’ to ‘men’ and ‘women’ respectively.

- Introduction, line 20-22: “In the present study, we adopted a brief BS integrated with routine TB appointment for smoking cessation....” Authors may consider revising this sentence.

Thanks for the suggestion. We have now rephrased the sentence as following:

“We adopted have previously developed, in collaboration with local teams in Bangladesh and Pakistan, a brief BS integrated with routine TB appointment for smoking cessation.[7] In the present study, over-and-above the BS, we examined the effectiveness and cost-effectiveness of the relatively low cost pharmacotherapy cytisine for smoking cessation in TB patients.[8]”

- Discussion, page 11, line 1: “Lack of any clear underlying hypotheses or evidence, subgroup analyses by patient characteristics or deterministic sensitivity analysis of key parameters were not planned..”. Authors may consider revising this sentence.

Thanks for the suggestion. We have now changed the sentences to the following:

“Subgroup analyses by patient characteristics and deterministic sensitivity analysis of key parameters were not planned because of the lack of clear underlying hypotheses. Moreover, limited by the research capacity, the sample size of the subgroups was likely to be insufficient to produce valid results.”

- Supporting information_1 (detailed methods and results of secondary analyses: “Upon further investigation, there were more participants who had hospital stays in Pakistan than in Bangladesh in both arms”. In addition, Figure 2 has two values with only 2 decimals instead of 3.

The sentence has been rephrased to:

“Upon further investigation, more participants had hospital stays in Pakistan than in Bangladesh, regardless of which arm they were in.”

Decimals in Figure 2 have been corrected.

VERSION 3 – REVIEW

REVIEWER	van den Brand, Floor Maastricht University CAPHRI School for Public Health and Primary Care, Family Medicine
-----------------	--

REVIEW RETURNED	26-Jul-2022
-------------

GENERAL COMMENTS	Third review bmjopen-2021-049644 Thanks to the authors for clarifying why they did not choose a societal perspective for their analysis. I understand this decision and I think this choice is understandable and up to them. But in response to the argumentation of the authors I have some last remarks: 1. There are standardized methods to estimate productivity-loss without needing to know all the details of the production loss that the authors describe (for example the friction cost method) 2. Health-related quality of life is certainly the recommended outcome for economic evaluations from a societal perspective. 3. The authors' argument shows why an analysis from the societal perspective is relevant. A broad economic evaluation from a societal perspective including all costs, such as productivity costs, could have shown (if cytisine had been effective) policy makers and employers that cytisine is cost-effective and convince them to invest in this intervention.
---